# The Role of Chronological Age, Health, and Basic Psychological Needs for Older Adults' Travel Intention

**Sangguk Kang** [1]**, Chen-Kuo Pai** [1] **and Donghan Kim** [2,*]

[1]   Faculty of Hospitality and Tourism Management, Macau University of Science and Technology,
      Macao SAR, China; ksangguk@must.edu.mo (S.K.); ckpai@must.edu.mo (C.-K.P.)

[2]   Colleage of Hotel and Tourism Management, KyungHee University, Seoul 02447, Korea

*    Correspondence: kimdh@khu.ac.kr

**Abstract:** This study examined how demographic information, chronological age, older adults' physical and mental health, and basic psychological needs affected travel intention. The survey samples were collected from 577 adults, all over 60 years of age. A hierarchical multiple regression analysis was used to test the proposed hypotheses. First, demographic information with chronological age was used for primary analysis. The outcome indicated that chronological age was negatively associated with travel intention while all other demographic variables were not. Secondly, when physical and mental health condition variables were added, physical health positively affected travel intention while chronological age still negatively affected travel intention. Thirdly, psychological needs (autonomy, competence, relatedness) for travel were included in the final analysis. The outcome showed that all psychological needs variables had a significant impact on travel intention for those with a physical health condition. However, chronological age was not a significant factor in travel intention during this analysis. This study shows that chronological age is not always an important factor that affects older adults' travel intention when other health and psychological variables are considered. This study provides some practical implications and tips for travel industry managers who are targeting the aging population.

**Keywords:** chronological age; physical health; mental health; competence; autonomy; relatedness; older adults; travel

## 1. Introduction

According to data by The World Bank [1], life expectancy has increased from 70 years old in 1960 to 79 years old in 2014. Considering the characteristics of tourism (e.g., the tourism industry is highly affected by low and high season), as stable and constant travel product consumers, the aging population not only plays an important role in offsetting risk but can also be seen as a highly profitable market. Senior citizens are becoming a more powerful spending group in the travel industry as the Baby Boomer generation increases in age. Since out-of-home activities are more positively connected with higher levels of happiness and well-being than in-home activities [2,3], travel might facilitate a more active lifestyle and improved personal development, which may contribute to better physical and psychological health in the aging population [4,5]. However, as chronological age is accompanied by many individual and social changes such as the deterioration of one's physical condition, retirement, and the loss of a spouse, these meaningful markers of transition can all affect older adults' travel behavior [5,6]. With this trend, tourism marketers have made extra efforts through customized services (e.g., professional help services) to attract older travelers for more successful business [7].

On the other hand, as older adults are living longer due to sufficient nutrition and better health education, chronological age differences might not be the only factors which affect older adults' travel behavior. Since life-expectancy has increased during the last several decades, various demographic variables including education, gender, and income, have also shown improvements in older adults' individual physical and mental health status. In addition, since older adults more frequently encounter physical dysfunction, chronical illness, and loss of loved ones (i.e., friends), older adults might have less competency, autonomy and relatedness which can cause changes in their travel behavior [8,9]. According to self-determination theory (SDT) by Deci and Ryan [10], satisfied basic needs are important to promote psychological well-being and these needs are not only universal across cultures but important in all aspects of a person's life. Although average longevity of life has increased almost eight years in the last six decades, the definition of 'old' varies throughout different agencies, organizations and scholars.

In the tourism field, many tourism scholars have also used the term 'older adults' arbitrarily, without deep consideration, in terms of research that shows that travel behavior of older adults is different amongst researchers and different age categories. Although research on travel behavior of older adults has been conducted to figure out which factors are critically important, those factors are still blurred and arguable due to extended life expectancy. For these reasons, it is essential to identify how these conditions affect older adults' travel behavior since chronological age might not be the only proxy of older adults' behavioral determination. Therefore, this study examined how various demographic information, physical and mental health condition, and basic psychological needs (i.e., competence, autonomy, relatedness) affect older adults travel intention.

## 2. Literature Review

### 2.1. Behavioral Change by Aging, Health and Other Demographic Status

As aging is associated with various possible changes such as physical dysfunction, mental health, and social relationships throughout different points in the lifespan [11–13], various fields including health, vocation, social work, travel, and leisure have studied older adults as a separate group from other younger generations (e.g., Hubley and Russell [14]). For instance, Jönsson and Devonish [15] conducted travel behavior studies with visitors to the Caribbean island of Barbados. The authors categorized age into three groups (18 to 35 years, 36 to 55 years, 56 years and over) and one-way ANOVAs were applied. Results showed that travel behavior varies throughout different age groups even at the same travel destination. Musick and Wilson [16] examined relationships between volunteering and depression based on two different age groups (less than 65 years and over 65 years). Authors used data from Americans' Changing Lives (ACL) studies (1986, 1989, 1994) and the outcome showed that those aged less than 65 years old had no statistical relationship between volunteering and depression while those aged 65 years or older saw a decrease in depression through volunteering. This result showed that different age groups have different behavioral outcomes.

On the other hand, Horneman et al. [17] profiled older travelers (age 60 years and over) from an Australian perspective and age was categorized by three groups (under 65, 65–74, 75 and over). Outcomes showed that age was only meaningfully correlated with pioneers and big spenders, and older-old respondents were less interested in seeking new experiences compared to younger-old ones. In addition, those older adults who received a lower level of education showed high preferences with package travel while those with a higher level of education preferred new experiences in their travel. This result showed no gender differentiation. Regarding travel motivation, outcomes showed that older adults gave highest priority to their health condition, followed by spending time with family and friends, visiting the place where they always wanted to go, and escaping from routine life. From this study, Horneman and colleagues [17] strongly disagreed that age is not the only determining factor for older adults' travel

behavior; other factors (i.e., characteristics, education level) should be considered. Pinquart [18] clarified the relationship between subjective health and objective health and argued that "age is not the cause of change in subjective health in old age but rather a proxy of age-associated declines in objective health and functional status" (p. 415). de Frias and Whyne [19] recruited 134 community-dwelling adults (50–85 years of age, *M* = 65.43, SD = 9.50) from Dallas, TX, to identify stress on health-related issues with quality of life. Outcomes showed psychological states such as life stress and mindfulness were significant factors for health conditions while age, gender, and education level were not statistically significant on physical or mental health. This study showed that age is not the main factor for affecting the health condition in an aging population.

On the other hand, as subjective health ratings embody overall features of physical well-being and inexpensive methods which can measure older adults' travel behavior, tourism scholars have used perceived health conditions as proxy of older adult's individual health status [4,20]. For instance, according to research by Jang and Wu [4], self-perceived health status proved statistically significant in the push motivation factors (i.e., knowledge-seeking, self-esteem). This outcome shows that older adults who are physically healthier are more internally motivated and possibly travel more often than unhealthy older adults. Regarding demographic status, Gibson and colleagues [21] conducted a study on a sport tourist market based on life-span perspective with three age groups (17–39 years, 40–59 years, 60–91 years) and asked 1277 New England residents to identify the trend of sport tourism participants (i.e., men vs. women, characteristics of three age groups). From the results, although sport tourism and age had a negative relationship for both men and women among the 60–91 years group, the pattern of sport tourism participation was not continuously decreased by age (i.e., men had less participation during middle age but increased later life while women had more participation in middle age than older age) and other environmental factors such as health condition, and level of education affected individual participation. Backman, Backman and Silverberg [22] conducted nature-oriented senior travelers aged 55 years old and over. Backman and colleagues [22] divided older adults into two groups and found that older senior travelers (over 65 years old) have more preferences to spend more time at one travel destination, visiting relatives and friends than younger old travelers (55–64). Jang and Wu [4] surveyed 60 years old or over Taiwanese older adults who were studying at continuing education classes at 13 senior centers and traveled at least one night a year either domestically or internationally prior to the survey. From the outcomes, health status had both a significantly positive and negative affection the older adults' travel behavior.

Although research by Jang and Wu had no age difference in travel behavior, Backman and colleagues [22] research showed age difference is an important factor for travel behavior. Huang & Tsai [20] surveyed older travelers and showed almost half of the subjects marked their health as "well" while the other half of the respondents were either "few problem" or "sick and handicapped". This study divided age into three-subgroups (55–59, 60–64, 65 and over). From the results, although age did not have at statistically significant impact on travel behavior, good health was a more significant factor on travel behavior than poor health. As health condition might be different even in the aging population amongst different age groups, health condition might play an important role in determining older adults' travel behavior. Since age is highly intertwined with a decrease of older adults' health condition and change of demographic status, examining health condition and other demographic status factors are critical for understanding older adults travel behavior [23]. In this context, the hypotheses drawn is as follows:

**H1.** *Demographic status including age, education, income, gender, and employment have significant impact on older adults' travel intention.*

**H2.** *Physical and mental health status have significant impact on older adults' travel intention.*

## 2.2. Basic Psychological Needs for Travel as Determinant Factors for Older Adults' Travel Behavior

As a powerful valid proxy of behavior, behavioral intention is closely linked to the engagement of actual behavior. Travel intention can be more a dominant proxy for travel behavior since older travelers are relatively freer from social duty (i.e., work, family care), and have more disposable income and time than younger generations [24,25]. Travel intention can be a powerful proxy of behavior since travel needs to be planned ahead through complex psychological stages, unlike daily product consumption. Basic psychological needs are considered as an initial condition for individuals' behavior for psychological growth and well-being and these basic needs posit as innate desire and are universal with a similar level of biological needs (e.g., hunger, sleep; Deci and Ryan [10]). From basic psychological needs, autonomy refers to the experience that one can freely choose activities and make decisions in accordance with one's goal. Competence refers to the desire to control behavior and master one's environment of activity which is similar concept with self-efficacy. The competence also comprises of the ability to manage a challenging task [10]. The concept of 'needs' was studied by various fields since needs play a fundamental role in human behavior [26]. For instance, Springer et al. [27] conducted qualitative studies to verify which basic psychological needs are critical for long-term physical activity adherence. From the in-depth interviews, participants were asked to explain their physical activity adoption and adherence based on basic psychological needs themes and found all of competence, autonomy, and relatedness to be considered important for continuous physical activity. These results are also meaningful as travel requires physical movement from one's home to one's travel destination. Although travel is one of the self-chosen, voluntary, and intrinsic interests-based activities, older adults might lose their autonomy and competence as their age is affected by physical and psychological changes. Relatedness refers to a sense of either connection to others or belongingness [10]. More specifically, relatedness is the feeling of being associated with others who are also connected to caring for others or cared for by others [28].

Aging is often accompanied by a weakening in physical and cognitive function as well as the narrowing of social relationships [29]. Since older adults enter a stage of transition in their social status by retiring, having continuous social relationships with other similar age groups might be different. Therefore, these three basic psychological needs are essential for individual behavior. In other words, when individuals have not met these three needs, an inherent tendency for well-being might be weakened. Hsu et al. [30] conducted in-depth interview to find out motivation of senior adults from both Beijing and Shanghai areas in China. They interviewed 12 females and 15 males from various age groups ranging from 55 to 90 years old. Among 27 interviewees, 13 of those older adults were under 70 years old, 11 older adults were in the 70s, two older adults were in their 80s, and one older adult was 90 years old. With this context, Hsu et al. proposed their own model consisted of 'external conditions' and 'internal desires'. External conditions were comprised with personal financial resources, improved living conditions, time resources, and health conditions. Internal desires showed seven motivational sub-themes: improving mental and physical wellbeing, escaping from daily routine, socializing with people, pride and patriotism, reward for hard work, and nostalgia. Although the study was conducted with highly limited samples in China, the outcome showed how older adults' travel behavior can be varied based on individuals' experienced environment. From the previous research, although basic psychological needs are an innate state and occur over a lifespan, it can be differentiated by various factors such as age, environment, and personal traits. Therefore, the basic psychological needs (i.e., autonomy, competence, and relatedness) were used to predict older adults travel intention [10].

**H3.** *Basic psychological needs for travel including autonomy, competence, and relatedness have significant impact on older adults' travel intention.*

## 3. Method

### 3.1. Sample Selection and Data Collection

The survey was collected summertime in 2017, using a paper-based survey. A total of 600 surveys were distributed in various areas (i.e., fast food restaurants, parks, and aging community centers) in Midwestern areas in the USA utilizing a convenience sampling method. First author visited parks, fast food restaurants, and YMCA for the data collection. Most of the data collection was conducted in the morning because many older adults frequently visit data collection sites in the morning. A total of 23 responses were excluded due to main part of survey items were not completed. Therefore, 577 surveys used for this study. A small gift (a bookmark) was provided as compensation to the survey participants. This research was approved by Indiana University (IRB #1611236441), USA.

### 3.2. Research Design and Structure

The self-administered questionnaires were developed through comprehensive literature reviews from travel, leisure, and health fields. The proposed study used a cross-sectional design with an onsite paper survey. As life expectancy was 79 years old in 2017 in US, this study delimited to participants who are above 60 years old. Therefore, respondents were only limited at least 60 years old and over to be included in the study. This study asked general free and pleasurable travel not by occupational or business travel. The paper-based survey questionnaire consisted of three parts: (1) the introduction of the study for subject consent, (2) the main study survey questionnaires: personal health condition, basic psychological needs for travel, travel intention, and (3) demographics information. More specifically, the first part of the survey was designed to introduce the study purpose to gain participants' consent, and to screen out ineligible respondents who younger than 60 years old. In the second part of the survey, respondents were asked to answer questionnaires from both theoretical and practical measurement scale. Lastly, in the third part of the survey, respondents were answered demographic information (i.e., gender, age, education, marital status).

### 3.3. Instruments

#### 3.3.1. Physical and Mental Health Condition

Current health status was measured with physical and mental health condition since chronological age is accompanied with both physical and mental health decrease [23]. Both physical and mental health status items were derived from health status questionnaire (SF-12, 36) items which is widely used in health-related studies (e.g., Brazier and Roberts [31]; Machicado et al. [32]; Stewart et al. [33]). Physical and mental health condition scale with two items were examined with five-point Likert scale of 1 = 'None of time', 5 = 'All of the time' and analyzed with reverse coding. The mean scores of each item showed that physical health (M = 3.95) and mental health (M = 3.98).

#### 3.3.2. Basic Psychological Needs for Travel

Basic psychological needs (nine items) were adapted and modified version of travel needs satisfaction scale was used to measure the basic psychological needs [34]. In this study, this construct was named as 'basic psychological needs for travel' and respondents were asked to rate their level of basic psychological needs for travel on the nine items listed below to measure this concept. To measure this concept, respondents were asked to rate their level of basic psychological needs for travel on the nine items which are consisted of autonomy (three items), competence (three items), and relatedness (three items). The participants marked how true they perceive their needs for travel scale, on a five-point likers scale of 1 = 'Not at all

true', 5 = 'Extremely true.' Cronbach alpha was examined to check the reliability of model construct and all constructs showed over 0.8 which is higher than recommended minimum level of 0.7 [35]. Table 1 shows the detail information of mean scores of basic psychological needs for travel.

**Table 1.** Descriptive Statistics for Basic Psychological Needs for Travel Scale (*n* = 577).

| Measurement Items | M | SD | Cronbach Alpha |
|---|---|---|---|
| **Physical health condition** | | | |
| During the last 4 weeks, how much of the time has your physical health interfered with your social activities, like visiting with friends, relatives, etc.? | 3.95 | 0.863 | |
| **Mental health condition** | | | |
| During the last 4 weeks, how much of the time has your emotional problems (ex: down hearted, stress, depression) interfered with your social activities, like visiting with friends, relatives, etc.? | 3.98 | 0.898 | |
| **Basic psychological needs for travel** | | | |
| **Autonomy** | | | |
| If I am making a travel decision now, I feel that my choice expresses my "true self." | 3.75 | 0.740 | |
| I feel like I am free to decide for myself where and how to travel. | 3.79 | 0.796 | 0.965 |
| Travel reflects my true interests and values. | 3.77 | 0.778 | |
| **Competence** | | | |
| Travel leads me to feel that I can take on and master hard challenges. | 3.63 | 0.771 | |
| Traveling makes me feel that I can successfully complete difficult tasks. | 3.65 | 0.758 | 0.930 |
| I am capable of handling travel. | 3.74 | 0.845 | |
| **Relatedness** | | | |
| I feel a strong sense of intimacy with the people I spend time within travel. | 3.62 | 0.826 | |
| Through travel, I expect to be connected with people who are important to me. | 3.60 | 0.808 | 0.974 |
| Travel could help me get closer to people who care for me, and whom I care for. | 3.60 | 0.833 | |
| **Travel intention** | | | |
| Whenever I have a chance to travel, I will travel. | 3.75 | 0.904 | 0.939 |
| I will do my best to improve my ability to travel. | 3.77 | 0.847 | |
| I will keep on gathering travel-related information in the future. | 3.82 | 0.843 | |

### 3.3.3. Travel Intention

Travel intention was adopted by Lee at al. [36] and three items were used in this study. Five-point Likert-type scale was used and the response categories of the scale ranged from 1 = 'very unlikely' to 5 = 'very likely'. Table 1 shows the detail information of mean scores of travel intention scale.

### 3.3.4. Control Variables

In this study, several demographic items were controlled in the hierarchical multiple regression models since age, income level, employ status, education, and health status are significantly associated with older adults' travel behavior.

### *3.4. Data Analysis*

Descriptive statistics and correlations analysis were executed to summarize collected sample information and checked the relationships the variables. Hierarchical multiple regression analysis was performed to examine the relationship between independent variables (e.g., demographic information, health condition, basic psychological needs for travel) and travel intention. In the first step, demographic information including age, gender, education level, household income status, employ status were analyzed to predict travel intention. In the second step, physical and mental health status were added to predict

travel intention. In the third step, basic psychological needs for travel were added to predict travel intention. All the analysis was performed using the SPSS 25 software package.

## 4. Results

First, descriptive statistics were analyzed to define the characteristics of the respondents. Demographic information of the samples in this main test is shown in Table 2. Respondents were aged 60–90 (M = 69.59, SD = 6.25) and a little more than half were male. The largest age group in the sample was located between 60–69 (53.6%). The most respondents were Caucasian (88.6%) and about 60 percentage of samples earned a higher degree with either bachelor's or graduate degrees. In 2016, respondents with income range ($30,000 to $49,999) showed highest (41.1%) and travel more than 100 miles showed highest (35.7%) with four to six times. Around 12.7% showed more than 10 times. Most respondents (58.8%) reported an employment status with retired from work. Second, the mean scores of each item showed that physical health (M = 3.95) and mental health (M = 3.98). These results show that older adults have both good mental and physical health condition. Regarding basic psychological needs, Cronbach's $\alpha$ was 0.966 with autonomy, 0.930 with competence, and 0.974 with relatedness which is high internal consistency. With travel intention, Cronbach' $\alpha$ was = 0.939. Third, the Pearson correlation coefficient matrix showed significant statistical relationships between demographic information, health condition, and basic psychological needs for travel.

**Table 2.** Demographic Information.

| Characteristics | Frequency | Percentage (%) |
|---|---|---|
| Gender | | |
| Male | 281 | 48.7 |
| Female | 296 | 51.3 |
| Age | | |
| 60–69 | 309 | 53.6 |
| 70–79 | 231 | 40.1 |
| 80 and over | 37 | 6.3 |
| Employment | | |
| Full Time | 120 | 20.8 |
| Part Time | 118 | 20.5 |
| Retired | 339 | 58.8 |
| Education | | |
| High School | 236 | 40.9 |
| College | 228 | 39.5 |
| Graduate School | 113 | 19.6 |
| Ethnicity | | |
| Caucasian | 511 | 88.6 |
| African American | 53 | 9.2 |
| Asian | 13 | 2.3 |
| Marital Status | | |
| Married | 294 | 51 |
| Cohabitating | 22 | 3.8 |
| Single, widowed, divorced | 261 | 45.2 |
| Household income 2016 | | |
| Less than $14,999 | 10 | 1.7 |
| $15,000–$29,999 | 80 | 13.9 |
| $30,000–$49,999 | 237 | 41.1 |
| $50,000–$74,999 | 128 | 22.2 |
| $75,000 or More | 122 | 21.1 |
| Travel experience in 2016 | | |
| None | 12 | 2.1 |
| Less than 3 times | 142 | 24.6 |
| 4–6 times | 206 | 35.7 |
| 7–9 times | 144 | 25 |
| 10 times or more | 73 | 12.7 |

Table 3 shows that chronological age was negatively related to travel intention ($r = -0.187$. $p < 0.01$) while there were no significant relationships between other demographic information (i.e., gender, education level, employment status, income level) and travel intention. As expected, travel intention was positively related to both physical ($r = 0.331$, $p < 0.01$) and mental ($r = 0.291$, $p < 0.01$) health conditions. Among basic psychological needs for travel, competence and relatedness ($r = 0.267$, $p < 0.01$) showed significant positive relationship to travel intention, following autonomy ($r = 0.233$, $p < 0.01$). Regarding the relationships between age and health condition, both physical ($r = -0.322$, $p < 0.01$) and mental ($r = -0.323$, $p < 0.01$) health condition showed significant negative relationship to chronological age.

**Table 3.** Pearson Correlations.

|      | TI | AG | GE | ED | EM | IN | PH | MH | AU | CO | RE |
|------|------|------|------|------|------|------|------|------|------|------|------|
| TI   | 1 | | | | | | | | | | |
| AG   | −0.187 ** | 1 | | | | | | | | | |
| GE   | −0.020 | −0.021 | 1 | | | | | | | | |
| ED   | 0.027 | −0.129 ** | 0.206 ** | 1 | | | | | | | |
| EM   | −0.029 | 0.287 ** | 0.054 | 0.025 | 1 | | | | | | |
| IN   | 0.042 | −0.211 ** | 0.086 * | 0.691 ** | −0.084 * | 1 | | | | | |
| PH   | 0.331 ** | −0.322 ** | 0.004 | −0.006 | −0.048 | 0.123 ** | 1 | | | | |
| MH   | 0.291 ** | −0.323 ** | 0.030 | −0.006 | −0.045 | 0.125 ** | 0.836 ** | 1 | | | |
| AU   | 0.233 ** | −0.188 ** | −0.018 | 0.144 ** | −0.035 | 0.200 ** | 0.296 ** | 0.262 ** | 1 | | |
| CO   | 0.267 ** | −0.173 ** | −0.023 | 0.121 ** | −0.005 | 0.145 ** | 0.313 ** | 0.250 ** | 0.361 ** | 1 | |
| RE   | 0.267 ** | −0.212 ** | 0.010 | 0.099 * | −0.121 ** | 0.119 ** | 0.232 ** | 0.158 ** | 0.304 ** | 0.306 ** | 1 |

Note. * $p < 0.05$, ** $p < 0.01$. TI = travel intention. AG = age. GE = gender. ED = education. EM = employ status. IN = income in 2016. PH = physical health condition. MH = mental health condition. AU = autonomy. CO = competence. RE = relatedness.

Third, hierarchical multiple regression analysis was examined to identify the effect of demographic information including chronological age, physical and mental health condition, and basic psychological needs for travel on travel intention. Table 4 shows the summarized results.

**Table 4.** Summary of Hierarchical Multiple Regression Results.

| Variable | Model 1 | | | Model 2 | | | Model 3 | | |
|------|------|------|------|------|------|------|------|------|------|
|  | *B* | *SE* | *β* | *B* | *SE* | *β* | *B* | *SE* | *β* |
| Constant | 5.529 | 0.434 | | 3.522 | 0.500 | | 3.574 | 0.488 | |
| Age | −0.025 | 0.006 | −0.195 *** | −0.012 | 0.006 | −0.092 ** | −0.008 | 0.006 | −0.059 |
| Gender | −0.044 | 0.069 | −0.027 | −0.053 | 0.066 | −0.033 | −0.037 | 0.064 | −0.023 |
| Education | 0.004 | 0.040 | 0.005 | 0.043 | 0.039 | 0.062 | 0.023 | 0.038 | 0.033 |
| Employ | 0.029 | 0.044 | 0.029 | 0.009 | 0.042 | 0.008 | 0.016 | 0.041 | 0.016 |
| Income | 0.002 | 0.046 | 0.002 | −0.044 | 0.045 | −0.055 | −0.055 | 0.044 | −0.069 |
| Physical health | | | | 0.261 | 0.068 | 0.276 *** | 0.163 | 0.068 | 0.173 ** |
| Mental health | | | | 0.036 | 0.066 | 0.039 | 0.059 | 0.064 | 0.065 |
| Autonomy | | | | | | | 0.060 | 0.035 | 0.073 * |
| Competence | | | | | | | 0.097 | 0.035 | 0.119 ** |
| Relatedness | | | | | | | 0.125 | 0.034 | 0.153 *** |
| $R^2$ | | 0.036 | | | 0.120 | | | 0.175 | |
| $F$ | | 4.30 * | | | 26.90 ** | | | 12.58 ** | |
| $\Delta R^2$ | | | | | 0.083 ** | | | 0.055 ** | |

Note. * $p < 0.01$. ** $p < 0.05$. *** $p < 0.001$. Outcome Variable: Travel Intention.

At the step 1 (H1), results showed that only chronological age (β = −0.195, $p < 0.001$) was negatively associated with travel intention. All other control variables such as gender, education level, employ status, and income level were statistically unrelated to travel intention. At the step 2 (H2), physical and mental health condition were added and results showed that physical health condition (β = 0.276, $p < 0.001$) was positively associated with travel intention while mental health condition was not statistically associated with travel intention. Chronological age (β = −0.092, $p < 0.05$) was still negatively associated with travel intention. At the step 3 (H3), basic psychological needs for travel were added. Among three variables, relatedness (β = 0.153, $p < 0.001$), competence (β = 0.119, $p < 0.05$), and autonomy (β = 0.073, $p < 0.01$) were positively significant on travel intention. Besides basic psychological needs for travel, physical health condition (β = 0.173, $p < 0.05$) was still positively significant to travel intention while chronological age was not statically significant to travel intention at this stage. The final model $R^2$ was 0.175 ($p < 0.001$) with whole variables which explain 17.5% of the variance in travel intention.

## 5. Conclusions

Although the aging population has become an increasingly prominent topic in tourism research, exploring the important indicators for older adults' travel behavior is still at an infant stage. Therefore, the present study examined which predictable variables (i.e., demographic information including chronological age, physical and mental health, basic psychological needs for travel) are associated with older adults' travel intention. The results from the hierarchical multiple regression analysis showed that only chronological age is negatively significant on travel intention while other demographic variables (i.e., gender, income, education) were not significant on travel intention at the first analysis stage. Although age had negative influence on travel intention, the variance was not large. This result supports that there are other factors which affect individual behavior since chronological age is accompanied with other individual and social changes such as physical or mental health condition, psychological state change, and loss of meaningful ones [5,6]. In this regard, as a second step analysis, physical and mental health condition was included in the model, and only physical health was positively significant on travel intention. At this stage, chronological age was still negatively associated with travel intention but negative chronological age $\beta$ was decreased. Other demographic variables and mental health were not significant on travel intention. From the results, there was significant change of variance and negative chronological age $\beta$ was decreased. These changes explain that physical health condition is an important predictor for travel intention since travel frequently requires healthy physical state. In addition, in this model, chronological age was less negatively significant to travel intention which shows that there are more other predictors can make change standardized regression coefficient of chronological age on travel intention. When basic psychological needs for travel (competence, autonomy, relatedness) variables were added in the model, physical health and all basic psychological needs for travel variables were significant while chronological age was not anymore significant on travel intention. At this stage, although physical health showed highest significant on travel intention, physical health $\beta$ was decreased compared to step 2. Among basic psychological needs for travel variables, relatedness showed highest significant on travel intention following competence and autonomy. From the results, there was significant change of variance. These outcomes are meaningful since there are other predictors (physical health condition, basic psychological needs for travel) statistically significant on travel intention and chronological age is not significant on travel intention. Although chronological age is strongly related with negative behavior in aging population, this analysis showed that chronological age is not meaningful when basic psychological needs variables are considered. As de Frias and Whyne [19] found, although stress and mindfulness were significant on health-related quality of life while age, gender, and education level were not statistically significant, this analysis showed when other psychological states controlled, chronological age was not significant on behavioral intention.

## 6. Discussion

From the results of the current study, it clearly indicates how older adults' basic psychological needs and physical health condition are strongly influencing their travel intention. In addition, the results suggest that chronological age is not consistent when other variables were considered together. Therefore, predictions of older adults' travel behavior can be misjudged and prejudiced when based solely on chronological age. More studies that look at the relationship between physical health condition and basic psychological needs as well as other factors on travel intention should be conducted in the future. Studies such as these can avoid misjudgment of older adults' travel behavior.

## 7. Limitations and Suggestions for Future Research

A first limitation is that because this study was only conducted using a sample of mid-western areas in the USA, it is difficult to generalize the results to other populations. Individuals in different environments (e.g., city, local, countryside) might have different concerns when it comes to travel intention. A second limitation is that this study did not include travel preference which might influence older adults' current and future travel behavior. Since this study is aimed at older adults over 60 years of age, they might have their own travel identity that is either negative or positive toward travel behavior. However, this study did not examine the potential influence of the emotional aspects on travel intention (negative or positive toward travel later in life) that could highly influence older adults' travel behavior. Third, the definition of travel was somewhat unclear and only limited to travel more than 100 miles one way or travel outside of a residential area with a one-night stay. If older adults are only interested in day travel, then the outcomes might be different since they exclude excursion as part of their travel. Since this study is not aimed at certain travel programs, more clear travel terms need to be defined, such as excursion, domestic, or international travel. All in all, future studies should include more samples from a variety of age groups and different environments as well as more predictors such as travel preference and a clear definition of travel.

**Author Contributions:** Conceptualization, S.K.; Formal analysis, S.K. and D.K.; Methodology, S.K., D.K. and C.-K.P.; Writing—original draft, S.K.; Writing—review & editing, C.-K.P. and D.K.; Super vision, S.K.; Validation, D.K.; Project administration, S.K.

**Funding:** This research received no external funding.

**Conflicts of Interest:** The authors declare no conflict of interest.

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
