# Peer review of "The Role of Chronological Age, Health, and Basic Psychological Needs for Older Adults’ Travel Intention"

_sustainability, doi:10.3390/su11236864_

Round 1

Reviewer 1 Report

There are so many grammatical errors in the paper it is difficult to read. You must hire a copy editor and never submit a paper that is this poorly written to a journal until it is cleaned up and 99% grammatically correct. Aside from that, there are flaws in the types of references cited -- for example on page 1 when discussing travel and personal development you cite a study of Taiwanese elders as if that is applicable to all older travelers. Given that your population was largely American you have to be careful when generalizing research from one cultural population to another as we know there are vast differences in motivations, expectations and experiences between travelers from different countries and cultures. 

Some of your conclusions just do not make sense to the reader about the significance and impact of chronological age on travel decisions -- you state when mental health was factored in, chronological age was negatively affected and when psychological factors were included (in the same realm as mental considerations in some ways) chronological age was not a factor. The way these statements are worded in the abstract and throughout the paper is very confusing. Also, with such a large corpus of literature on travel among older populations, it seems to be getting too far off track spending time citing articles about volunteering and depression when you could include materials closer to your area of study.

The subject is fine -- examining travel motivations and experiences among older populations and which variables are most influential, but this article is such a grammatical and logical mess that it just needs to be scrapped or re-worked thoroughly. As it stands now, it is nowhere near ready to publish.  

Author Response

There are so many grammatical errors in the paper it is difficult to read. You must hire a copy editor and never submit a paper that is this poorly written to a journal until it is cleaned up and 99% grammatically correct.

- Based on your tip, asked to edit a document from a professional academic editor.

Aside from that, there are flaws in the types of references cited -- for example on page 1 when discussing travel and personal development you cite a study of Taiwanese elders as if that is applicable to all older travelers. Given that your population was largely American you have to be careful when generalizing research from one cultural population to another as we know there are vast differences in motivations, expectations, and experiences between travelers from different countries and cultures.

- Based on your comment, we added more supportive samples to American. This research is in the reference number 5.  

Ahn, Y. J., & Janke, M. C. (2011). Motivations and benefits of the travel experiences of older adults. Educational Gerontology, 37(8), 653-673.

Some of your conclusions just do not make sense to the reader about the significance and impact of chronological age on travel decisions -- you state when mental health was factored in, chronological age was negatively affected and when psychological factors were included (in the same realm as mental considerations in some ways) chronological age was not a factor. The way these statements are worded in the abstract and throughout the paper is very confusing.

- Thank you for the comment regarding mental health conditions and psychological factors. Mental health condition is about emotional problems such as depression, stress. However basic psychological needs are autonomy, competence, and relatedness. Here, autonomy explains that one can freely choose activities and make decisions in accordance with one’s goal. Competence refers to the desire to control behavior and master one's environment of the activity which is a similar concept with self-efficacy. Relatedness refers to a sense of either connected to others or belongingness.  

It looks similar to word, but the concept of meaning is different.

Also, with such a large corpus of literature on travel among older populations, it seems to be getting too far off track spending time citing articles about volunteering and depression when you could include materials closer to your area of study.

- We understand the reviewer's concern since this study is for older adults’ travel behavior. However, to explain older adults' behavioral change, we brought other filed cases to identify how aging makes different emotional conditions. As you see from the literature review, we also explained these behavioral changes with older adults’ travel-related studies.

Reviewer 2 Report

The manuscript “The Role of Chronological Age, Health, and Basic Psychological Needs for Older Adults` Travel Behavior” analyses the impact of socio-demographic variables and of health conditions on travel intention of a defined sample of older individuals. This is a very interesting research!

In general, I think this is a well-written manuscript, clear, with little English writing/grammar errors.  In general, the discussion of your results could be improved. However I suggest to revise some aspects:

1) I think that the title could be modified as: “The Role of Chronological Age, Health conditions, and Basic Psychological Needs for Older Adults` Travel intention”. In this way I think that title could be clearer and linked to the content.

2) abstract:

In general: I think it is important to specify which types of travel are involved (tourism, cults, health, etc.)

line 6: the survey sample information were …

line 7: delete from first - to- analysis.

Line 12: positively correlated with or health positively affected travel intention…

Lines 13-14: please rewrite. There is a verbal problem. Also at line 17

3) introduction

For my opinion, we should add data on travel flows (differentiating the types - tourist / religious / etc.) - in particular related to the individuals target considered in the research.

Line 66 ANOVAs were applied?

Line 67: results showed: please standardize verbs form

4) M&M

Line 205: Delete the dot after “all of time”.

Line 205- 206. It is a result

I think it is necessary to explain better the questions used to evaluate  the physical and mental health condition. Also in case of the 9 items used for psychological needs evaluation.

Lines 218-220: it is a result

Table 1 reported results related to likert scale …I think that this table could be moved in the result section.

In addition, in this section a table reported the items analysed is necessary.

5) results

Table 3 title position

6) the conclusion section it is Discussion and conclusions

However, there is little discussion of your results. The manuscript discussion needs to be improved and supported by literature references.

Author Response

The manuscript “The Role of Chronological Age, Health, and Basic Psychological Needs for Older Adults` Travel Behavior” analyses the impact of socio-demographic variables and of

health conditions on travel intention of a defined sample of older individuals. This is a very interesting research! In general, I think this is a well-written manuscript, clear, with little English writing/grammar errors.

Thank you for your positive comment about this research. Based on your advice, we asked to edit a document from professional academic editor.

In general, the discussion of your results could be improved. However I suggest to revise some aspects:

1) I think that the title could be modified as: “The Role of Chronological Age, Health conditions, and Basic Psychological Needs for Older Adults` Travel intention”. In this

way I think that title could be clearer and linked to the content.

- Based on your comment, we decided to change our title as you recommended.

2) abstract:

In general: I think it is important to specify which types of travel are involved (tourism, cults, health, etc.)

- We have not categorized this study with a specific type of travel. However, we will specify the type of travel for the next research.

line 6: the survey sample information were …

- Based on your comment, we have revised it.

line 7: delete from first - to- analysis.

- First, demographic information with chronological age was used for primary analysis. Regarding hierarchical multiple regression, we conducted demographic information including chronological age as a first step.

Line 12: positively correlated with or health positively affected travel intention…

- Based on your comment, we have revised it.

Lines 13-14: please rewrite. There is a verbal problem. Also at line 17

Thirdly, when psychological needs (autonomy, competence, relatedness) for travel added for analysis, all psychological needs variables were significant to travel intention with physical health. However, chronological age was not significant in travel intention. This study indicates that chronological age is not always an important factor that affects older adults' travel intention when other variables considered.

- Based on your comment, we have revised it like below.

Thirdly, psychological needs (autonomy, competence, relatedness) for travel were included in the final analysis. The outcome showed that all psychological needs variables had a significant impact on travel intention for those with a physical health condition. However, chronological age was not a significant factor in travel intention during this analysis. This study shows that chronological age is not always an important factor that affects older adults' travel intention when other health and psychological variables are considered.

3) introduction

For my opinion, we should add data on travel flows

(differentiating the types - tourist / religious / etc.) – in particular related to the individuals target considered in the research.

- This research addressed older adults’ travel intention in general. With that, in the literature review part, we introduced different types of tours and individual targets with the aging population.

Line 66 ANOVAs were applied?

- Based on your comment, we have revised it from examined to applied.

Line 67: results showed: please standardize verbs form

- Based on your comment, we have revised it from shows to show.

4) M&M

Line 205: Delete the dot after “all of time”.

- Based on your comment, we have deleted the dot after “all of time”

Line 205- 206. It is a result

- Based on your comment, we have moved to the result part. You can see line 247-249.

I think it is necessary to explain better the questions used to evaluate the physical and mental health condition. Also in case of the 9 items used for psychological needs evaluation.

- Table 1 shows all the question items information. 

Lines 218-220: it is a result

- Based on your comment, we have moved to the result part. You can see line 249-251.

Table 1 reported results related to likert scale …I think that this table could be moved in the result section.

In addition, in this section a table reported the items analysed is necessary.

- Since Table 1 shows all the question items we put this table right after the instrument.

5) results

Table 3 title position

-- Based on your comment, we have adjusted the position of table 3.

6) the conclusion section it is Discussion and conclusions

However, there is little discussion of your results. The

manuscript discussion needs to be improved and supported

by literature references.

-- Based on your comment, we have relocated and adjusted the discussion part. You can see the line from 323 – 331.

Reviewer 3 Report

            The presented paper is valuable attempt to examine how demographic information, chronological age, older adults’ physical and mental health, and basic psychological needs affect travel intentions.

               Among the weaknesses of the presented text - in my opinion - should be indicated:

1. no explanation regarding the chosen method of selecting the research sample;

There is no explanation regarding the chosen method of selecting the research sample. What were the reasons for choosing it? The advantages and disadvantages of this choice should be indicated. Furthermore, can the results be considered as representative? (N=577)?

2. lack of clear information on the verification of research hypotheses;It is worth presenting information about the acceptance (rejection) of research hypotheses.

3. no comparison to other studies;

In addition, the Conclusions part should emphasize what new the results of the research have brought, and it is necessary to supplement this point with a comparison with the results of other authors.

            In my opinion, the article will be suitable for publication, however, after completing the above information and answers to the above questions.

Author Response

The presented paper is valuable attempt to examine

how demographic information, chronological age, older adults’

physical and mental health, and basic psychological needs

affect travel intentions.

- Thank you for your positive comment about this research. 

Among the weaknesses of the presented text - in my opinion - should be indicated:

no explanation regarding the chosen method of selecting the research sample;

There is no explanation regarding the chosen method of selecting the research sample.

- Based on your comment, we added “utilizing convenience sampling method” under Line 183.

What were the reasons for choosing it? The advantages and disadvantages of this choice

should be indicated. Furthermore, can the results be considered as representative? (N=577)?

- This research only recruited older adults over 60 years old. As life expectancy was 79 years old in 2017 in the US, this study delimited to participants who are above 60 years old. This study asked general free and pleasurable travel, not by occupational or business travel. 577 samples can be difficult to represent all older adults. However, with limited time and resources, that was the most samples we could take. In addition, compared to other research related to older travelers, the sample size is more than the other research.

lack of clear information on the verification of research hypotheses; It is worth presenting information about the acceptance (rejection) of research hypotheses.

- Based on your comment, we added H1, H2, and H3 after all the step1, 2, and 3. This can show clear information on the verification of research hypotheses and information about independent variables on the dependent variable.

no comparison to other studies; In addition, the Conclusions part should emphasize what new the results of the research have brought, and it is necessary to supplement this point with a comparison with the results of other authors.

- Based on you and other reviewer comments, we made a discussion part to address new results from our study.  From the discussion, we addressed that “From the results of the current study, it clearly indicates that how older adults’ basic psychological needs and physical health condition are strongly influencing travel intention. In addition, the results suggest that chronological age is not consistent when other variables were considered together. Therefore, it can be misjudged and prejudiced when predicting older adult's travel behavior only based on chronological age. Although age is not the consistent predictor on travel intention, physical health condition and basic psychological needs for travel strongly related to travel intention, more future study recommended with other variables to avoid misjudgment of older adults’ travel behavior.” You can see this from line 323 to 331.

In my opinion, the article will be suitable for publication, however, after completing the above information and answers to the above questions.

- Thank you for your positive comment about this study again and we have tried to update our study based on your valuable comment. 

Round 2

Reviewer 1 Report

There is a very long passage between line 67 and 135 with no paragraph structure -- you need to break this up more.

I would still like to know more about how the survey was administered -- were any researchers on site with the paper surveys that were placed at different locations? Were they just left there? Were there people at the organizations and places facilitating in any way? Was this survey approved through Human Subjects Review as health information was being collected? 

Line 336-- your editor missed a few items - this is not grammatically correct: "are strongly influencing on travel intention.." Ditto on line 338: "Therefore, it can be misjudged and prejudiced when predict older adults travel  behavior only based on chronological age." 

I think you will need to rework this sentence too, line 339: "Although age is not the consistent predictor on travel intention, physical health condition and basic psychological needs for travel strongly related to travel intention, more future study recommended with other variables to avoid misjudgment of older 341 adults’ travel behavior." 

This whole section starting on line 343 is also riddled with awkward sentences and grammatical issues -- please do not submit materials to journals with this degree of error: 

"Limitations and Suggestions for Future Research 
A first limitation is that because this study was only conducted using a sample of mid-western areas in USA, it is difficult to generalize the results to other population. Individuals in different environment (e.g., city, local, countryside) might have different confrontation on travel intention. Second limitation is that this study did not include travel preference which might influence on older adults’ current and future travel behavior. Since this study is aimed older adults over 60 years old, they might have their own travel identity as negative or positive on travel behavior. However, this study could not examine the potential influence of emotional (negative or positive on travel in the later life) aspect on travel intention which might highly influence on older adults’ travel behavior. Third, the definition of travel was somewhat not clear and only limited with travel more than 100 miles one way or traveled outside of resided area with one-night stay. If older adults are only interested day travel, then the outcomes might different since they exclude excursion as a travel. Since this study is not aimed certain travel programs, more clear travel terms need such as domestic or international travel. All in all, future studies should need more samples with various age stage and different environment as well as more predictors such as travel preference with clear travel definition"

Author Response

There is a very long passage between line 67 and 135 with no paragraph structure -- you need to break this up more. I would still like to know more about how the survey was administered -- were any researchers on-site with the paper surveys that were placed at different locations?

First, the author visited parks, fast food restaurants, and the YMCA for data collection. Most of the data collection was conducted in the morning because many older adults frequently visit data collection sites in the morning. Added inline from 186-188.

Was this survey approved through Human Subjects Review as health information was being collected?

This research was approved by Indiana University, Bloomington, USA.

 IRB STUDY #1611236441 including health information questions.

Line 336-- your editor missed a few items - this is not grammatically correct: "are strongly influencing on travel intention.." Ditto on line 338: "Therefore, it can be misjudged and prejudiced when predict older adults travel behavior only based on chronological age."

I think you will need to rework this sentence too, line 339: "Although age is not the consistent predictor on travel intention, physical health condition and basic psychological needs for travel strongly related to travel intention, more future study recommended with other variables to avoid misjudgment of older 341 adults’ travel behavior."

Discussion

From the results of the current study, it clearly indicates how older adults’ basic psychological needs and physical health condition are strongly influencing their travel intention. In addition, the results suggest that chronological age is not consistent when other variables were added and considered together. Therefore, predictions of older adults travel behavior can be misjudged and prejudiced when based solely on chronological age. More studies that look at the relationship between physical health condition and basic psychological needs as well as other factors on travel intention should be conducted in the future. Studies such as these can avoid misjudgment of older adults’ travel behavior.

This whole section starting on line 343 is also riddled with awkward sentences and grammatical issues - Changed as you see below

Limitations and Suggestions for Future Research

A first limitation is that because this study was only conducted using a sample of mid-western areas in the USA, it is difficult to generalize the results to other populations. Individuals in different environments (e.g., city, local, countryside) might have different concerns when it comes to travel intention. A second limitation is that this study did not include travel preference which might influence older adults’ current and future travel behavior. Since this study is aimed at older adults over 60 years of age, they might have their own travel identity that is either negative or positive toward travel behavior. However, this study did not examine the potential influence of the emotional aspects on travel intention (negative or positive toward travel later in life) that could highly influence older adults’ travel behavior. Third, the definition of travel was somewhat unclear and only limited to travel more than 100 miles one way or travel outside of a residential area with a one-night stay. If older adults are only interested in day travel, then the outcomes might be different since they exclude excursion as part of their travel. Since this study is not aimed at certain travel programs, more clear travel terms need to be defined, such as excursion, domestic, or international travel. All in all, future studies should include more samples from a variety of age groups and different environments as well as more predictors such as travel preference and a clear definition of travel.

More information is attached by word file.